# Immune Microenvironment in Childhood Cancers: Characteristics and Therapeutic Challenges

**DOI:** 10.3390/cancers16122201

**Published:** 2024-06-12

**Authors:** Anup Singh Pathania

**Affiliations:** Department of Biochemistry and Molecular Biology, The Fred and Pamela Buffett Cancer Center, University of Nebraska Medical Center, Omaha, NE 68198, USA; anup.pathania@unmc.edu

**Keywords:** pediatric cancers, immune microenvironment, tumor-immune biology, immune suppression, immunotherapy, metastasis

## Abstract

**Simple Summary:**

The immune microenvironment surrounding a growing tumor is complex and diverse. Understanding this complexity is crucial for predicting prognosis and response to cancer therapy. Various immune cells, signaling molecules, and extracellular matrix components within this microenvironment interact intricately, influencing tumor progression and treatment effectiveness. A deeper understanding of these interactions can lead to better prognostic markers and more effective, personalized therapeutic strategies. Compared to adults, the pediatric tumor immune microenvironment is less studied. Research on different pediatric tumors has shown that regulatory signaling networks between tumors and surrounding immune cells are critical in shifting the balance between pro- and anti-tumor immune responses. Therefore, gaining insights into the tumor immune composition is key to improving the efficacy of various pediatric cancer therapies, particularly immunotherapies that enhance host anti-tumor immunity.

**Abstract:**

The tumor immune microenvironment is pivotal in cancer initiation, advancement, and regulation. Its molecular and cellular composition is critical throughout the disease, as it can influence the balance between suppressive and cytotoxic immune responses within the tumor’s vicinity. Studies on the tumor immune microenvironment have enriched our understanding of the intricate interplay between tumors and their immunological surroundings in various human cancers. These studies illuminate the role of significant components of the immune microenvironment, which have not been extensively explored in pediatric tumors before and may influence the responsiveness or resistance to therapeutic agents. Our deepening understanding of the pediatric tumor immune microenvironment is helping to overcome challenges related to the effectiveness of existing therapeutic strategies, including immunotherapies. Although in the early stages, targeted therapies that modulate the tumor immune microenvironment of pediatric solid tumors hold promise for improved outcomes. Focusing on various aspects of tumor immune biology in pediatric patients presents a therapeutic opportunity that could improve treatment outcomes. This review offers a comprehensive examination of recent literature concerning profiling the immune microenvironment in various pediatric tumors. It seeks to condense research findings on characterizing the immune microenvironment in pediatric tumors and its impact on tumor development, metastasis, and response to therapeutic modalities. It covers the immune microenvironment’s role in tumor development, interactions with tumor cells, and its impact on the tumor’s response to immunotherapy. The review also discusses challenges targeting the immune microenvironment for pediatric cancer therapies.

## 1. Introduction

Despite cancer being considered a rare disease in children due to its low prevalence, it is the second leading cause of death in children aged 1 to 14. According to the World Health Organization (WHO), an estimated 400,000 children and adolescents aged 0 to 19 years develop cancer each year. Among this demographic, blood cancers reign as the most prevalent, with leukemias comprising 36.1% of cases in children aged 0–4 years and 15.4% in those aged 15–19 years, while lymphomas account for 5.3% and 22.5%, respectively, in the same age groups [1,2]. Following leukemia, central nervous system (CNS) tumors make up roughly 17.2% of cases in children aged 0–4 years, 26.3% in those aged 5–9 years, and 20.0% in those aged 10–14 years.

Subsequently, sympathetic nervous system tumors like neuroblastoma are most common in the 0–4 years age group (12.5%), followed by renal tumors (8.9% in 0–4 years but less prevalent in 15–19 years), and bone tumors (4.7% in children aged 0–14 years, and 7.8% in those aged 15–19 years). Adolescents aged 15–19 years exhibit a heightened incidence of epithelial tumors, melanoma, and lymphoma [1]. Data from United States (US) Cancer Statistics spanning from 2003 to 2019 show that the overall cancer incidence rate among children and adolescents under 20 years old was 178.3 per 1 million, with the highest rates observed in leukemia (46.6), CNS neoplasms (30.8), and lymphoma (27.3) [3].

Despite an overall increase in pediatric cancer incidence, the death rate has declined by more than half from 1970 to 2021 and remains significantly lower than that of adults with cancer [4,5]. According to WHO’s 2020 report, the most frequently diagnosed adult cancers globally are breast (2.26 million cases), lung (2.21 million cases), colon and rectum (1.93 million cases), and prostate (1.41 million cases) [6]. The majority of cancer-related deaths are attributed to lung (1.80 million), colon and rectum (916,000), liver (830,000), stomach (769,000), and breast (685,000) [6]. In the US, breast, lung, prostate, and colorectal cancers comprise almost 50% of new cases, while lung, colorectal, pancreatic, and breast cancers make up nearly 50% of cancer-related deaths [5].

Pediatric cancers differ significantly from adult cancers, possessing distinct characteristics rather than simply being miniature versions of adult tumors. A comprehensive pan-cancer DNA study on approximately 1700 pediatric cancer patients, including those with acute lymphoblastic leukemia (ALL), acute myeloid leukemia (AML), neuroblastoma, Wilms (kidney) tumor, and osteosarcoma, revealed significant differences in somatic DNA alterations between childhood and adult tumors [7]. Compared to adult tumors, childhood tumors exhibit a lower mutational burden and commonly show somatic DNA alterations, such as deletions, duplications, single-nucleotide variants (SNVs), or rearrangements, which are more specific to the particular histotype. Additionally, the report highlights the identification of 142 driver genes in pediatric cancers, of which only 45% overlap with those found in adult pan-cancer studies [7].

Contrary to adult tumors, which exhibit multiple-driver genetic alterations, pediatric cancers are enriched with single-driver mutations [8]. These mutations are more specific to individual cancer types and show minimal overlap across different pediatric cancers [7,8]. However, the presence of inherited germline mutations is less than 10% in both children and adolescents with cancer [8,9]. Like adult tumors, many pediatric ALL, neuroblastoma, and osteosarcoma, show increased mutations in their relapsed state, making them resistant to anticancer drugs [10,11,12]. This suggests that selective mutational pressure plays a significant role in their survival and conferring resistance to the drugs.

Advancements in DNA sequencing technologies in recent years have greatly contributed to our understanding of both solid and hematological malignancies in children. Comprehensive studies employing whole-genome and exome sequencing have uncovered notable differences in driver genes affecting essential cell proliferation pathways like RAS, Janus kinase/signal transducer and activator of transcription (JAK-STAT), and Phosphoinositide 3-Kinases (PI3K) signaling among pediatric cancer populations [7]. For example, somatic alterations in genes like anaplastic lymphoma kinase (ALK), neurofibromatosis type 1(NF1), and phosphatase and tensin homolog (PTEN) primarily contribute to pathway deregulation in solid tumors, whereas in leukemias, mutations in fms-like tyrosine kinase 3 (FLT3), phosphatidylinositol-4,5-bisphosphate 3-kinase, catalytic subunit alpha (PIK3CA) or p110α, phosphoinositide-3-kinase regulatory subunit 1 (PIK3R1), and RAS genes are more common.

Additionally, specific mutations exhibit varying prevalence between solid and blood cancers, highlighting their distinct genetic profiles [7]. For instance, two KRAS isoforms are common in leukemias, accounting for approximately 70% of cases, whereas they are infrequent in solid tumors. Additionally, somatic alterations in cyclin-dependent kinase inhibitor 2A (CDKN2A), particularly deletions, are highly prevalent in T-cell ALL (T-ALL) and B-cell ALL (B-ALL) cases, occurring in 78% and 42%, respectively, in contrast to solid tumors where they are present in 11% or less of cases [7].

Further distinctions between pediatric solid and hematological malignancies are evident in the composition of immune cells within the tumor microenvironment (TME). Various immune cell populations, including tumor-infiltrating lymphocytes (TILs), regulatory T cells (Tregs), natural killer (NK) cells, B cells, macrophages, dendritic cells (DCs), and myeloid-derived suppressor cells (MDSCs), play crucial roles in the development and progression of these cancers. The proportions of these immune cells within the TME influence anti-tumor immunity and impact various tumor characteristics. Additionally, the TME encompasses tumor stromal cells, such as fibroblasts, endothelial cells, pericytes, and extracellular matrix (ECM) components like collagen, laminin, and fibronectin, along with non-coding RNAs, growth factors, cytokines, and chemokines, all of which further influence tumor behavior.

In comparison to solid tumors, hematological cancers develop and progress within lymphoid organs where immune cells reside and often initiate anti-tumor immune responses. This close association with immune cells primarily arises from deficiencies in antigen priming [13]. Conversely, in many solid tumors, T-cell priming typically occurs but is impeded at the tumor site by diverse immune suppressive mechanisms. However, recent studies on pediatric AML and B-ALL reveal that these cancers contain infiltrating activated cytotoxic T cells capable of generating neoantigen-specific CD8+ T-cell responses, yet they suffer from chronic exhaustion and lack phenotypes associated with stem-like memory cells [14]. This differs from solid brain and CNS tumors, which are considered immune cold, characterized by high myeloid signatures and low T-cell infiltration [15,16,17]. This distinction is further underscored in a recent study, where an analysis of the immunogenomic and transcriptomic profiles of 202 pediatric tumor cases, focusing on relapsed or refractory solid tumors, uncovered a significant lack of immune infiltration across different tumor types [18].

Similar to solid tumors, pediatric AML blasts increase the number of MDSCs, drive macrophages toward a pro-tumoral phenotype, and exhibit signs of exhausted NK cells [14,19]. The correlation of NK cell infiltration and activation with enhanced outcomes in pediatric leukemia is well-documented in studies, thus sparking considerable interest in NK cell-based immunotherapies for this ailment [20]. With the advent of immunotherapy, analyzing the immune microenvironment to comprehend the interplay between tumors and adjacent immune cells has garnered substantial attention in recent decades. Various types of immunotherapies including monoclonal antibodies (mAbs), transfer of immune cells from donor to patient, checkpoint inhibitors, and vaccines have been tested in different solid tumors. Unlike traditional cancer treatments such as surgery, chemotherapy, or radiation therapy that directly target cancer cells, immunotherapy aims to stimulate the immune system’s natural ability to detect and attack abnormal cancer cells in the body. Immunotherapy has shown remarkable success in shrinking tumors and improving survival rates in some adult cancer patients with various tumor types [21].

Given its promising outcomes, numerous studies have aimed to characterize the immune microenvironment of pediatric tumors. While much research has focused on adult tumors, there is a growing emphasis on profiling the immune landscape surrounding pediatric cancers. The success of Chimeric Antigen Receptor (CAR) T-cell therapy, particularly in targeting the CD19 antigen in B-ALL, holds promise for enhancing immune responses against childhood tumors [22].

However, the immune microenvironment of pediatric tumors differs from that of adult tumors in several ways. The immune system in children is less developed because it is still maturing and exhibits distinct histological and molecular features compared to adults. Children rely more on innate immunity, which is generally non-specific, rather than specific adaptive immunity. In the context of the TME, pediatric tumors have a lower mutational burden and different genetic alterations compared to adult tumors, which can result in unique antigen profiles and immune interactions within the pediatric TME. Additionally, the absence of mutation-derived antigen expression makes these tumors less immunogenic, allowing them to evade or subvert anti-tumor immunity and develop acquired resistance to immunotherapy. Consequently, pediatric tumor patients show different outcomes when treated with drugs studied in adult tumors. To achieve better and improved outcomes, a proper understanding of tumors and their microenvironment in pediatric populations of different ages is crucial for effective cancer therapy.

This review aims to provide a comprehensive overview of recent literature on profiling the immune microenvironment of different pediatric tumors. It summarizes findings on the characterization of the immune microenvironment in pediatric tumors and its role in influencing the TME, interactions between the immune system and tumor cells, tumor growth and development, metastasis, and the response of tumors to immunotherapy. Additionally, the review discusses the substantial challenges associated with targeting the immune microenvironment for future therapies in pediatric cancers.

## 2. Profiling the Immune Microenvironment of Pediatric Tumors

Many studies have shown that densities of tumor-infiltrating immune cells, including cytotoxic lymphocytes, MDSC, tumor-associated macrophages (TAM), and mast cells in the pediatric TME are strongly associated with immune suppression and anti-tumor immune escape. The TME can be classified into different categories based on the presence and distribution of T lymphocytes. TME with infiltrated immune cells is referred to as immune-inflamed, whereas if T cells are confined to the periphery of the tumor and unable to effectively penetrate the tumor mass, it is called immune-excluded. An “immune-desert” TME indicates the complete absence of T cells.

Molecular profiling of the immune microenvironment of high-risk pediatric tumors based on Immune Pediatric Signature Score (IPASS), which combines immunohistochemistry staining data of T lymphocytes with RNA and whole-genome sequencing, estimated that 31% of high-risk cancers in pediatric patients have infiltrating T cells [23]. In CNS tumors, pediatric gliomas have the most T-cell receptor (TCR) clones, while osteosarcoma and neuroblastoma show more diversity in these clones. Having a diverse range of TCR clones suggests a strong immune response [24]. This diversity in TCR repertoires is linked to prognosis and response to immune therapy in these tumors. Interestingly, the study found that tumor mutational burden (TMB) and neoantigen load, which reflect genetic changes in cancer cells, did not predict T-cell infiltration in pediatric cancers [23]. This implies that the presence of a high TMB or a large number of neoantigens may not necessarily correlate with increased T-cell infiltration and immune response in these cancers.

Similar findings were observed in a murine-based tumor progression glioma model that mimics the progression from low-grade gliomas (LGG) to high-grade gliomas (HGG) [25]. LGG tumors demonstrated a high level of T-cell homing, infiltration, and secretion of interferon-gamma (IFNγ), which is an activation marker for T cells, in comparison to HGG tumors. Furthermore, the study found a significant difference in the proportions of cytotoxic CD8+ and CD4+ T cells between the LGG and HGG tumor cores. LGG tumors exhibited a higher number of CD8+ T cells compared to HGG tumors. However, both LGG and HGG tumors showed relatively small proportions of CD4+ T cells. Moreover, the TME of LGG and HGG tumors harbors distinct macrophage clusters exhibiting different phenotypes [25]. These macrophages transition from an immune-activated phenotype in LGG to an immunosuppressive state in HGG, suggesting their role in immune evasion during different phases of tumor development.

The ability of TAMs to adopt various functional phenotypes is a critical factor in pediatric tumors, influencing tumor growth, the anti-tumor immune response, and therapy outcomes. For example, chemotherapy-treated MYCN-amplified neuroblastoma patients showed increased tumor infiltration of HLA-DR+, CD11c+, CD68+, and CD14+ pro-tumorigenic macrophages that expressed immunosuppressive molecules such as Tim-3, B7-H3, and CD163 [26]. Treatment of these patient’s tumors with anti-colony stimulating factor 1 receptor (CSF1R), which impairs macrophage infiltration, prevents the regrowth of chemotherapy-resistant relapsed tumors [26]. This implies that tumor-infiltrated macrophages play a key role in therapeutic resistance to chemotherapy in neuroblastoma patients.

Similarly, in retinoblastoma, the microenvironment is enriched with pro-tumorigenic macrophages expressing the M2-type microglial marker Iba1+ [27]. The population of these TAMs increases after therapy [27]. Tumor cells reprogram TAMs to promote immune suppression and resistance to anti-tumor drugs. This is supported by studies demonstrating that macrophages cultured in a conditioned medium obtained from retinoblastoma cells exhibit a more pro-tumor M2 phenotype [27,28]. M2-type TAMs are associated with various other types of pediatric tumors, including osteosarcoma, rhabdomyosarcoma, leukemia, lymphoma, and brain tumors. These TAMs express M2-phenotype markers such as CD163, CD204, and CD206, which have been shown to prevent cytotoxic T lymphocytes (CTL) infiltration into the tumor core [29,30,31,32,33].

In pediatric TME, TAMs can interact with MDSCs and other immune cells, thereby influencing the overall immune response within the tumor. MDSCs are a heterogeneous population of immature myeloid cells that commonly arise in the bone marrow and are recruited to the tumor site. MDSCs can suppress immune responses and potentiate TAM-mediated immune suppression, promoting tumor development and growth [34]. Comparative transcriptomic studies suggest that the TME of osteosarcoma pulmonary metastatic sites is significantly enriched with populations of pro-tumor TAMs and polymorphonuclear (PMN) MDSCs compared to primary bone tumors [35]. These TAMs express programmed death ligand-1 (PD-L1) and colocalize with CD8+ lymphocytes at the interface of pulmonary metastatic sites. The expression of PD-L1 on TAMs leads to inhibitory signaling in activated T cells, which downregulates T-cell activity during immune responses and confines activated T cells to peripheral regions [35].

Furthermore, PMN MDSCs are a crucial microenvironment component in hematological malignancies like leukemia and lymphoma. The population of PMN MDSCs decreases in patients who respond well to therapy, indicating their potential as a biomarker for treatment response [34]. In comparison to their normal counterparts, children with ALL exhibit a higher number of both CD33+CD11b+ monocytic MDSCs (M-MDSCs) and PMN-MDSCs. This increase in MDSC populations is significantly upregulated after chemotherapy [36]. Similarly, an elevated population of PMN-MDSCs, identified as CD45+CD19-HLA-DR-CD11b+CD33+CD15+ granulocytic MDSCs (G-MDSCs), is observed in the peripheral blood and bone marrow of children with B-ALL, in comparison to age-matched healthy controls [37].

The studies in the murine model of T-ALL have revealed important insights into the dynamic and functional interactions between MDSCs and other immune cells within the TME [38]. One significant finding is that MDSCs play a role in maintaining the proliferation of immune suppressive CD4+CD8+ T cells in T-ALL-bearing mice. These CD4+CD8+ T cells, in turn, promote the expansion of CD11b+Gr-1+ MDSCs. This reciprocal interaction between MDSCs and CD4+CD8+ T cells contributes to the immune suppressive environment within the TME. Interestingly, treatment with an anti-interleukin 6 (IL6) antibody impedes the accumulation of intratumoral MDSCs, correlating with reduced percentages of CD4+CD8+ T cells [38].

Chemokines and cytokines are critical immune components in the pediatric TME that play an important role in regulating the local balance of pro-tumor or anti-tumor phenotypes. Selective targeting of cytokine/chemokine-cytokine/chemokine receptor signaling networks can promote anti-tumor immunity by altering tumor biological and immunological phenotypes [39]. Moreover, many pediatric neuroblastoma, Ewing sarcoma (ES), osteosarcoma, and ALL, show altered expression of circulating cytokines/chemokines such as CCL2, CXCL4, CXCL6, CXCL10, and CXCL12 [40,41,42]. These circulating chemokines might be clinically utilized as prognostic markers in these tumors.

## 3. Influence of the Immune Microenvironment on Pediatric Tumor Biology

Genetic and epigenetic alterations in pediatric tumors play an important role in fostering the pro-tumorigenic features of the tumor immune microenvironment. Altered genetic expression in tumor cells drives the expression of cytokines that play a critical role in the recruitment and phenotype of immune cells, especially cells of the myeloid lineage. Understanding these complex and dynamic interactions is essential for comprehending the mechanisms behind immune evasion strategies in pediatric cancers.

### 3.1. Immune Cell–Tumor Cell Interactions and Their Role in Tumor Progression

Single-cell RNA sequencing analysis of neuroblastoma tumors reveals distinct subtypes of immune cell populations, including myeloid, NK, B, and T cells [43]. These cells form a complex, dynamic, and heterogeneous ecosystem with tumor and stroma cell populations in the TME. Continuous interactions occur between membrane receptors expressed on immune cells and extracellular ligands present on tumor or stroma cells. These interactions play a significant role in shaping the TME ecosystem of pediatric tumors, including neuroblastoma, and can influence tumor growth as well as the response to therapy.

For example, neuroblastoma tumors exhibit elevated levels of the immune checkpoint molecule CD200, which interacts with its corresponding receptor CD200R [44]. CD200R is present on the surface of HLA-DR+CD14+ myeloid cells, CD11c+ dendritic cells, and CD4+ and CD8+ T cells. CD200 expression in neuroblastoma tumors plays a role in regulating the composition of these immune cells and impacting the anti-tumor immune response within TME. Neuroblastoma tumors with increased CD200 expression display reduced numbers of CD4+ and CD8+ T cells, which exhibit more differentiated phenotypes and produce lower amounts of effector and proliferation marker proteins such as IFN-γ and tumor necrosis factor-alpha (TNF-α) [44].

In pediatric AML and B-ALL, overexpression of CD200, along with other receptors such as CD123 and CD56, on leukemic cells and leukemia stem cells has been associated with adverse prognostic features and poorer clinical outcomes [45,46]. This suggests that CD200 expression in these cancers may contribute to treatment resistance and disease aggressiveness. The specific mechanisms by which CD200 expression impacts these cancers are linked to immune modulation and evasion, as seen in neuroblastoma. Upregulation of CD200 in AML cells is associated with an increased population of tumor-suppressive FoxP3+ Tregs, low T-cell cytokine production due to reduced MAPK and STAT3 signaling, impaired T-cell metabolism, decreased production of CD107a+CD8+ memory T cells, reduced TNFα, IL-2, and IFNγ production in CD4+ memory cells, and upregulation of the exhaustion marker TIM-3 in tumor-infiltrating CD4+ bone marrow T cells [47,48,49,50].

Overexpression of immune checkpoint molecules in the TME is an important mechanism in evading immune surveillance and destruction in pediatric tumors. The interaction of immune checkpoint molecules such as cytotoxic T-lymphocyte-associated protein 4 (CTLA-4), programmed cell death protein 1 (PD-1), or B7-H3 receptor with corresponding ligands on tumor cells inhibits their anti-tumor activities, leading to immune suppression. For instance, CTLA-4 overexpression by leukemic cells decreases the expression of CD80, a co-stimulatory molecule, on T cells required for proper T-cell stimulation and activation. CD80 downregulation impairs the ability of T cells to mount an effective immune response against the tumor cells [51]. Pediatric patients with B-ALL exhibit significantly elevated levels of serum CTLA-4 that correspond to the B-ALL cell population, suggesting its role as a marker of disease severity [52].

Furthermore, CTLA-4 upregulation in different intratumoral T-cell populations, such as CD4+CD25highFoxp3+ Tregs, CD4+, and CD8+ T cells, has been associated with abnormal cytokine signaling and poor prognosis in B-ALL and aggressive pediatric sarcoma patients [53,54]. CTLA-4, expressed in T cells, competes with another receptor, CD28, for binding to the proteins CD80 and CD86, which are expressed on the surface of antigen-presenting cells (APCs). CD28 provides a co-stimulatory signal necessary for T-cell activation. When CTLA-4 binds with CD80 and CD86, it decreases CD28 interactions with these proteins, leading to the inhibition of T-cell activation and the maintenance of immune homeostasis. The overexpression of CTLA-4 on Tregs helps to dampen anti-tumor immunity [55]. Moreover, the interaction of CTLA-4 on Tregs with CD80 and CD86 on DCs leads to the downregulation of these co-stimulatory molecules [56]. This results in the instability of contacts between Treg cells and DCs, which limits local Treg cell proliferation. Disruption of this feedback loop through CTLA-4 blockade can induce Treg cell hyper-proliferation [56]. This suggests that, following anti-CTLA4 treatment, Treg cells can continue to proliferate and limit the anti-tumor immune response. Therefore, the functions of CTLA-4 need to be carefully considered in the TME when targeting CTLA-4 in cancer therapy.

Certain pediatric solid tumors, including neuroblastoma, osteosarcoma, retinoblastoma, and rhabdomyosarcoma, display increased PD-L1 expression on their surfaces following treatment. This overexpression is strongly associated with tumor recurrence and a poor prognosis in patients [57,58,59,60]. PD-L1 interacts with PD-1 receptors on immune cells, helping the tumor evade the immune system [57]. Prolonged immunotherapy can elevate PD-1 expression in CD4+ and CD8+ T cells, leading to reduced infiltration of these cells into the tumor and decreased levels of activation markers such as IFNγ, TNFα, and granzyme B [61]. Similarly, chemotherapy can enhance PD-1 and PD-L1 expressions on both tumor-infiltrating inflammatory immune cells and tumor cells, as observed in retinoblastoma. Notably, before treatment, most retinoblastoma tumors exhibit low or negative PD-L1 expression [62].

The loss of function of the RB1 gene, a tumor suppressor gene implicated in retinoblastoma, is associated with increased PD-L1 expression. The RB protein, regulated by phosphorylation at specific sites, can suppress PD-L1 expression by modulating the NF-κB signaling pathway [63]. Phosphorylation of the RB protein at serine-249/threonine-252 (S249/T252) by CDK4/6 promotes its interaction with the NF-κB protein p65, resulting in the inhibition of the NF-κB pathway. Consequently, the loss of RB function, either through knocking down RB or using a CDK4/6 inhibitor, selectively upregulates NF-κB pathway genes, including PD-L1 [63]. These findings suggest that RB plays a crucial role in regulating the immune response within the TME by modulating the NF-κB-PD-L1 pathway. Furthermore, when compared to neuroblastoma and retinoblastoma, osteosarcoma’s TME, particularly in metastatic sites, exhibits significant expression of PD-L1 on tumor cells, macrophages, and T cells, which is upregulated after chemotherapy [60,64]. By combining chemotherapy with PD-L1 inhibition, the immune response can be reactivated, thereby enhancing the anti-tumor activity of T cells in osteosarcoma tumors [64]. These findings demonstrate that the PD-1/PD-L1 signaling pathway promotes the establishment of an immunosuppressive TME that facilitates malignant development.

Moreover, B7-H3 exhibits sustained surface expression on tumor cells, primarily detected in the tumor microenvironment of certain pediatric malignancies. Similar to PD-L1, B7-H3 can interact with T cells, and suppress their anti-tumor activity [65]. Immune profiling of pediatric brain tumor patient-derived orthotopic xenografts (PDOX), representing both low and high grades, consistently demonstrated the presence of B7-H3 and GD2 on tumor cell surfaces [66]. Similarly, B7-H3 has been found to be associated with poor anti-tumor immunity, therapy response, and prognosis in other pediatric tumor types such as AML, medulloblastoma, Wilms tumor, and rhabdomyosarcoma [67,68,69,70]. This suggests that B7-H3 may be involved in regulating the immune response within the pediatric brain tumor microenvironment. Figure 1 depicts the dynamic interplay between tumor and immune cells and their roles within the pediatric TME.

### 3.2. Metastasis and Immune Microenvironment

Metastasis is a crucial event in tumor progression as it enables cancer cells to escape from the primary tumor and establish secondary tumors or metastases in distant sites of the body. Metastasis is responsible for the majority of cancer-related deaths [71]. In pediatric TME, non-cancer cells, particularly immune cells, secrete factors that can induce ECM remodeling, creating a more permissive environment for cancer cell movement and invasion. For instance, M2-TAMs associated with Wilms tumors activate AKT/PI3K signaling in cancer cells, initiating cellular changes linked to epithelial–mesenchymal transition (EMT) in the tumor cells [72]. In osteosarcoma tumors, TAMs upregulate COX-2, MMP9, and STAT3 phosphorylation in cancer cells, which promotes their migration, invasion, and metastasis to the lungs [73].

In human osteosarcoma lung metastasis specimens, a subpopulation of CD163+ macrophages was found to express erythropoietin receptor (EPOR), CD206, CD163, and PD1, which are known to play significant roles in TAMs immune suppressive and tumor-promoting functions [74]. Interestingly, another study observed more TAM infiltration into patient tumors that showed reduced metastasis and improved survival in high-grade osteosarcoma. Furthermore, it has been observed that macrophage numbers tend to increase following chemotherapy treatment, which is associated with a good therapeutic response [75]. These findings contradict the general understanding of TAMs, which are believed to be tumor-promoting, and suggest that TAMs may have different roles and functions depending on the specific tumor context. Moreover, the induction of M2-type polarization of TAMs by neuroblastoma tumor cells upregulates CXCL2 secretion in TAMs, which enhances tumor cell invasion [76].

TME data from neuroblastoma patients suggest that there is a higher infiltration of TAMs at tumor metastatic sites compared to localized tumors. Furthermore, the expression of specific markers on TAMs, including CD33, CD16, IL6R, IL10, and FCGR3, has been found to have a strong correlation with the survival prediction of patients with metastatic MYCN-nonamplified neuroblastoma [77]. This suggests that macrophage polarization significantly impacts molecular features and therapy-associated outcomes of metastatic neuroblastomas. MDSCs are another subset of the immune cell population that contributes to remodeling the ECM in the pediatric TME. A study conducted on M3-9-M mice, which is a syngeneic orthotopic tumor model of rhabdomyosarcoma that metastasizes to the lungs and shares similarities with human metastatic rhabdomyosarcoma, suggests that MDSCs are the main recruited cell population to secondary organ sites during metastasis. These MDSCs express higher levels of immunosuppressive factors and play a crucial role in the formation of pre-metastatic niches in the lungs [78].

MDSCs can interact with TAMs, dendritic cells, and Tregs to facilitate tumor metastasis. Tregs have been reported to impair local immune cell activity to support metastasis at secondary sites [79]. In patients with primary metastatic ES, Tregs were found at significantly higher frequencies compared to patients with localized disease, suggesting that Tregs may play a role in the development of metastatic disease in ES [80]. Similarly, lower intratumoral cytotoxic CD8+ T cell and higher FOXP3+ Treg cell populations were observed in osteosarcoma patients diagnosed with primary metastasis [81]. These findings indicate the significance of the interplay between various immune cells, including MDSCs, TAMs, dendritic cells, CD8+ T cells, and Tregs, in influencing tumor metastasis. Understanding these complex interactions may aid in developing targeted therapies to counteract the immunosuppressive microenvironment and improve outcomes for pediatric patients with metastatic tumors.

### 3.3. Immune Cell-Derived Cytokines

Secretion of immune-derived cytokines and chemokines, stimulated by cancer cells, is another critical point of interaction between tumor cells and infiltrated immune cells. These signaling molecules create a complex communication network within the TME that influences various aspects of tumor progression in most pediatric tumors. Specific changes in cytokine levels in the serum, including interleukins and TNFα, have been linked to disease severity, tumor progression, and therapeutic response in children with ALL and AML [82,83]. Tregs and MDSCs are the key immune cells that progressively accumulate within the TME and contribute to immune-suppressive interleukins, leading to the inhibition of anti-tumor activity driven by T and NK cells. Data suggest that Tregs-derived IL-10 and IL-35, work together to promote CD8+ T-cell exhaustion in the TME [84]. Both IL-10 and IL-35 induce the transcriptional repressor BLIMP1, which is known to promote terminal differentiation in T and B cell lineages and the expression of inhibitory receptors like PD-1, LAG3, TIM3, TIGIT, and 2B4 in CD4+ and CD8+ TILs [84,85]. More specifically, IL-35 promotes inhibitory receptor induction on TILs, inhibiting the differentiation of T cells into TCM (central memory T cells), whereas IL-10 regulates cytokine production and effector functions of TILs [84]. IL-10 has been found elevated and associated with the pathogenesis of pediatric neuroblastoma [86], medulloblastoma [87], leukemia, and lymphomas [88,89,90].

Cytokine profiling of CD4+ and CD8+ T cells from children with ALL revealed a significant reduction in IL-2 and IFNγ-producing CD4+ and CD8+ T cells below the normal range, while IL-4 production in CD4+ and CD8+ T-cell subsets increased [91]. A recent similar study in children with ALL has identified a correlation between elevated blood cytokine levels and ALL, suggesting a link between early-life immune events, cytokine imbalances, and the risk of ALL. The study found that patients with ALL exhibit higher levels of a group of correlated cytokines, including IL-1β, IL-8, TNF-α, and VEGF, at birth compared to their healthy counterparts [92]. Similar cytokine profiling has been conducted in children with B-ALL, anaplastic lymphoma kinase-positive anaplastic large cell lymphoma, pediatric pilocytic astrocytomas, as well as children with cancer and febrile neutropenia [93,94,95,96]. These findings collectively suggest an intriguing association between cytokine levels and tumor development.

Furthermore, the dysregulation of serum inflammatory cytokines, such as IL-1β, IL-6, and TNF-α, has been observed in pediatric patients with major depressive disorder (MDD), and it is linked to depression in children with cancer [97]. Cytokine deregulation can arise in response to cancer and its treatments, resulting in the upregulation of inflammatory reactions. Excessive cytokine production induces chronic inflammation, fostering an environment that promotes the development and progression of cancer. Therefore, dysregulated cytokine signaling signatures can be informative biomarkers in pediatric cancer research and clinical management. These signatures could improve cancer detection, patient stratification, and personalized treatment approaches by providing insights into the immune response and its interaction with cancer cells. Table 1 discusses some key cytokines in the pediatric TME and their roles and potential as predictive biomarkers.

### 3.4. Immunotherapy Response and Predictors in Pediatric Tumors

Before immunotherapy became widely recognized and used in cancer treatment, surgery, radiation, and chemotherapy were standard treatment options for childhood cancers. Immunotherapy has shown remarkable success in some advanced-stage cancers, leading to complete remission and potential cures. Immunotherapy as a multimodal therapy with surgery, radiation, or chemotherapy has shown better survival outcomes in pediatric patients [155,156]. A range of cancer immunotherapy approaches has been tested in pediatric cancer patients, including the use of cytokines like IL-2 [157], adoptive therapies like CAR-T and NK cell therapies [158,159], the use of targeted mAbs [160], immune checkpoint inhibitors [161], or oncolytic viruses that selectively infect and destroy cancer cells [162]. Despite these advances, the effectiveness of immunotherapy is limited in high-risk pediatric cancers due to the absence of recognizable markers. As a result, targeted mAbs and adoptive T-cell therapies have shown particular promise in treating specific pediatric cancer types that express tumor-specific antigens, such as leukemia, neuroblastoma, and melanoma.

Results from two clinical trials by Locatelli et al. and Brown et al. of antibody blinatumomab (Blincyto) in children and young adults with relapsed B-ALL showed that blinatumomab is more effective than chemotherapy [160,163]. Blinatumomab is a CD3/CD19-directed bispecific T-cell engager molecule (BiTE) that brings CD19-expressing leukemia cells and CD3-positive cytotoxic T cells together to facilitate T-cell-mediated leukemia cell killing. In both trials, blinatumomab-treated patients had better outcomes and experienced fewer side effects than patients who received additional chemotherapy as consolidation therapy [160,163]. In the Locatelli et al. study (*n* = 108), patients treated with blinatumomab exhibited a 2-year event-free survival of 66.2% compared to 27.1% for those receiving chemotherapy. Moreover, blinatumomab showcased higher rates of minimal residual disease (MRD)-negative remission (90% vs. 54%) and a greater proportion of patients proceeded to hematopoietic stem cell transplantation (HSCT) (88.9% vs. 70.4%) [160].

In the Brown et al. study (*n* = 214), the 2-year disease-free and overall survival rates were 54.4% and 71.3%, respectively, compared to 39.0% and 58.4% with chemotherapy. Blinatumomab also exhibited superior rates of MRD negativity (75% vs. 32%) and a higher percentage of patients proceeding to HSCT (70% vs. 43%) compared to the chemotherapy group [163]. However, there are some differences in patient selection in both studies that may explain some of the discrepancies in their treatment outcomes. The Locatelli et al. trial involved patients under 18, randomized after completing three cycles of chemotherapy, with approximately half already achieving MRD negativity [160]. In contrast, Brown and co-authors enrolled participants up to age 30, randomizing them after one cycle of chemotherapy, with only around 25% showing MRD negativity [163]. In addition, the Locatelli et al. study suggests that patients with very early relapse benefit most from blinatumomab, while the Brown et al. study indicates limited activity in patients responding unfavorably to reinduction therapy, emphasizing the significant impact of patient selection on treatment outcomes [163].

The other most notable mAbs approved for pediatric cancer patients include Dinutuximab, an anti-GD2 mAb for the treatment of neuroblastoma [164]. Anti-GD2 mAb is specific to neuroblastoma cells, as they express the tumor-specific antigen GD2, a glycolipid antigen with restricted expression in normal nerve tissues [165]. Anti-GD2 monoclonal antibodies have been used in combination with various administration regimens, including chemotherapy, cytokines like IL-2 and granulocyte–macrophage colony-stimulating factor (GM-CSF), or allogeneic NK cells in neuroblastoma treatment. Several types of anti-GD2 antibodies have been explored for therapeutic purposes in neuroblastoma patients, including the human anti-mouse mAb antibody 3F8 [166], human/mouse chimeric mAbs like Ch14.18/SP2.0 (dinutuximab, Unituxin^®^, United Therapeutics, Durham, NC, USA) [167,168], Ch14.18/CHO (dinutuximab beta, Qarziba^®^, EUSA Pharma, Hemel Hempstea, UK) [169], and Ch14.18K332A (Provenance Biopharmaceuticals, Waltham, MA, USA; St. Jude Children’s Research Hospital, Memphis, TN, USA) [170], alongside humanized antibodies like hu14.18-IL2 developed by the National Cancer Institute (NCI) [171] and hu3F8 (Naxitamab, DANYELZA^®^, naxitamab-gqgk) [172] developed by Memorial Sloan Kettering Cancer Center, New York, NY, USA. 

In vitro findings indicate disparities in their binding to GD2, duration of effectiveness, and cytotoxic strength. For instance, hu3F8 exhibits stronger binding to GD2 compared to Ch14.18 but has a shorter half-life [173]. Ch14.18K332A demonstrates heightened antibody-dependent cellular cytotoxicity (ADCC) by NK cells and neutrophils, while Ch14.18 displays lower binding and cytotoxic efficacy but persists longer [174,175]. The impact of these characteristics on clinical outcomes awaits future testing. Approved anti-GD2 antibodies for high-risk neuroblastoma treatment in the US include naxitamab (Danyelza^®^) and dinutuximab (Unituxin^®^), while in Europe, dinutuximab beta (Qarziba^®^) is available.

Furthermore, multicenter trials have been either completed or are currently underway to compare regimens containing anti-GD2 with conventional treatments. For instance, one trial investigated the efficacy of anti-GD2 (murine 3F8) treatment alongside GM-CSF in children in remission from stage 4 neuroblastoma [176]. Patients were assigned to three regimens: regimen A received 3F8 alone, regimen B received 3F8 + intravenous GM-CSF + 13-cis-retinoic acid (CRA) post-stem-cell transplantation (SCT), and regimen C received 3F8 + subcutaneous GM-CSF + CRA. Additionally, some ultra-high-risk patients were treated with regimen C. Over a 5-year period, survival rates demonstrated improvement, with regimen C exhibiting the highest progression-free survival (62%) and overall survival (81%). Moreover, ultra-high-risk patients experienced enhanced survival with this treatment approach [176]. Similarly, a randomized phase III trial conducted by the Children’s Oncology Group (COG) showed that the combination of anti-GD2 ch14.18 (dinutuximab) with GM-CSF and IL-2 could improve survival rates among children with high-risk neuroblastoma [177]. However, in another phase 3 trial utilized ch14.18/CHO (dinutuximab beta) and isotretinoin as maintenance therapy with or without subcutaneous IL-2, adding IL-2 did not enhance outcomes and led to increased toxicity compared to dinutuximab beta alone [121]. The observed differences in these two IL-2 studies could originate from variations in treatment protocols, including the incorporation of GM-CSF alongside IL-2 and the subcutaneous delivery of IL-2 in the first study. The rationale for integrating GM-CSF and IL-2 with dinutuximab is to enhance antibody-dependent cellular cytotoxicity (ADCC), primarily by amplifying granulocyte and NK cell counts and activation, which constitute the primary mechanism of dinutuximab-induced anti-neuroblastoma immune reactions. Moreover, in both trials, immunotherapy showed greater efficacy in patients with minimal residual disease or those who responded well to induction therapy but did not achieve a complete response. This highlights the potential advantage of immunotherapy for individuals displaying favorable responses to initial treatments. However, it also underscores the pressing need for continued refinement of treatment regimens to enhance outcomes and minimize toxicity in patients with recurrent or progressive disease, even following various treatments.

Other strategies tested to augment the effect of anti-GD2 mAbs with various combination therapies in neuroblastoma include Ch14.18 + irinotecan + temozolomide + GM-CSF [178], Hu14.18K322A + induction chemotherapy [179], Hu14.18-IL2 + expanded NK cells (NCT02573896), Ch14.18 + temozolomide + expanded γδ T cells (NCT05400603), Anti-GD2 CARs [180] and many others. These diverse strategies underscore the multifaceted efforts aimed at improving outcomes for patients with neuroblastoma, particularly those at high risk or in remission. Despite the use of anti-GD2, many patients still relapsed. Continued research and clinical trials in this area are vital for advancing treatment options and ultimately improving survival rates and quality of life for children battling this challenging disease.

Other antibodies approved for human use against different protein targets include rituximab, an anti-CD20 mAb for ALL [181]; ipilimumab, an anti-CTLA-4 antibody for pediatric patients with advanced melanoma [182]; and pembrolizumab, an anti-PD1 antibody approved for specific subsets of pediatric patients [183]. A study investigated the effectiveness of rituximab in newly diagnosed pediatric ALL patients who tested positive for CD20. Rituximab, combined with chemotherapy, has shown success in treating adult ALL and has been well received. In this pediatric trial, twenty patients were randomly assigned to either receive rituximab with standard chemotherapy (Intervention group) or undergo standard chemotherapy alone (Control group). The addition of rituximab resulted in lower absolute blast count and peripheral blood MRD levels on day 8, suggesting potential benefits in pediatric ALL treatment [181]. In another study, rituximab was used to reduce allergic reactions associated with Polyethylene glycol (PEG)-asparaginase (pegaspargase), a commonly used chemotherapy agent. Despite a decrease in CD20-positive B cells, newly diagnosed pediatric ALL patients who received rituximab experienced a significant incidence of infusion reactions. Additionally, there were no significant differences observed in pegaspargase allergies, anti-pegaspargase antibodies, or MRD levels between the rituximab and control groups [184]. These findings contrast with the previous study; however, both studies had a small number of participants, indicating the need for further research. Additionally, in the later study, rituximab was administered alongside multimodal induction therapy, which could have already reduced the leukemia burden and decreased the MRD, potentially masking any rituximab effects.

Similarly, ipilimumab has been assessed in 45 pediatric patients across two clinical trials. The first trial, a dose-finding study involving 33 individuals aged 2 to 21 years with relapsed or refractory solid tumors [185], and the second, an open-label, single-arm trial with 12 adolescents aged 12 to under 18 years with unresectable stage III or IV melanoma [186]. In both trials, ipilimumab demonstrated efficacy in some melanoma patients and exhibited a safety profile comparable to that observed in adults. However, the second trial was prematurely halted due to slow accrual caused by enrollment challenges, primarily because this disease is exceedingly rare in children [186]. Despite its rarity, there is a pressing need for parallel studies comparing children and adult cohorts for surgical management and innovative treatment options. This approach will facilitate the implementation of standardized guidelines, ensuring that medical practices are tailored to the specific needs of pediatric patients while benefiting from insights gleaned from adult studies, thus enhancing overall care and outcomes across different age groups.

There are numerous ongoing trials investigating these antibodies, as well as antibodies targeting other molecules, in combination with various immunotherapies or chemotherapy, for various pediatric tumors (Table 2). Moreover, adoptive cell therapy including a CD19-targeting CAR T-cell immunotherapy has been approved for subsets of pediatric patients with advanced leukemia [187]. Despite immunotherapy becoming a viable treatment option for many pediatric cancer patients, it does not work for everyone [188]. The primary reason for this is that the success of immunotherapy depends on the patient’s endogenous immunity. Several factors can limit the effectiveness of immunotherapy, including a limited number of neoantigens for immunotherapies to target due to a lower mutational load, an immunosuppressive tumor microenvironment that lacks anti-tumor cytotoxic lymphocytes, low surface expression of MHC-I, allowing evasion from T-cell-mediated anti-tumor immunity, and tumor heterogeneity arising from the presence of several malignant subpopulations within a single tumor mass [8,189].

To address these limitations, novel innovative approaches and procedures, combined with a deep understanding of intra-tumor complexity within the patients, are needed. Molecular profiling of a patient’s tumors is a key strategy for overcoming these challenges and improving the decision-making process for targeted therapies, ultimately leading to higher cure rates in cancer patients. Identifying specific molecular markers reflecting germline and somatic mutations, epigenetic changes in DNA, chromosomal aberrations, and protein expression alterations can help clinicians accurately stratify tumors on a molecular level and predict treatment responses. It also involves utilizing computational tools to predict neoantigens in patients with low mutational loads and develop personalized cancer vaccines targeting these neoantigens to stimulate the immune system.

Furthermore, combining immunotherapy with other treatment modalities like targeted therapy, immune-checkpoint inhibitors, or radiation therapy has been proven effective in enhancing the immune system’s response by increasing the number of neoantigens and reducing tumor heterogeneity [196]. Therefore, the use of rationally designed combinatory therapy strategies is likely to have a significant impact on pediatric tumors, provided the tumors are molecularly well defined. However, a molecular profile may evolve over time due to the accrual of molecular abnormalities that arise during cancer progression and are influenced by the selective pressures of previous treatments. As a result, there is a requirement for innovative technologies to continuously monitor cancer genotypes in real time, enabling necessary adaptations in cancer therapy regimens. Moreover, as advancements continue in cell-manufacturing approaches, adoptive immune-cell therapies, such as autologous or allogeneic transplantation of T or NK cells, or genetically modified CAR-T cells expressing novel T-cell receptors or chimeric antigen receptors, have achieved success in providing enduring responses in certain cases. These strategies are particularly effective in tumors with low antigen presentation or those characterized by an immune-suppressive tumor microenvironment.

However, despite promising developments in novel approaches and procedures and the widespread implementation of immunotherapy in pediatric tumors, not all patients receive similar benefits from the treatment. In addition, higher off-target cytotoxicity in combination therapies, tumor cells’ ability to evade anti-tumor immunity, and challenges associated with adoptive cell therapies like inflammation, cytokine release syndrome (CRS), neurological toxicity, manufacturing barriers, and limited accessibility pose significant challenges for their use in the field of pediatric oncology [197].

## 4. Conclusions

Like adult tumors, pediatric tumors also rely on a complex TME that surrounds them and contributes significantly to their growth, development, and metastasis. However, unlike adult tumors, the pediatric immune microenvironment remains largely unexplored. One significant area that is poorly understood is the complex signaling network triggered by specific oncogenic drivers within tumor cells that educate and shape immune cells toward a pro-tumor phenotype. Given the extensive diversity in the immune microenvironment and the critical signaling pathways triggered by tumor cells across various tumor types and tissues, it is imperative to investigate how to navigate and comprehend this diversity. As our comprehension of immune–tumor interaction networks advances, we can classify the immune microenvironment in diverse pediatric tumors. This classification facilitates the recognition of vital immune cells and their related signaling mechanisms involved in suppressing anti-tumor immunity. Such insights have the potential to accelerate responses to established standard-of-care therapies and constitute a noteworthy area for future research. While currently approved U.S. Food and Drug Administration (FDA) immunotherapy treatments exhibit restricted efficacy, strategically targeting the immune microenvironment emerges as an appealing approach for cancer treatment. Encouraging preclinical investigations across diverse pediatric tumors has demonstrated the potential to leverage the immune microenvironment of tumors for therapeutic advantages [140,198,199,200].

## Figures and Tables

**Figure 1 cancers-16-02201-f001:**
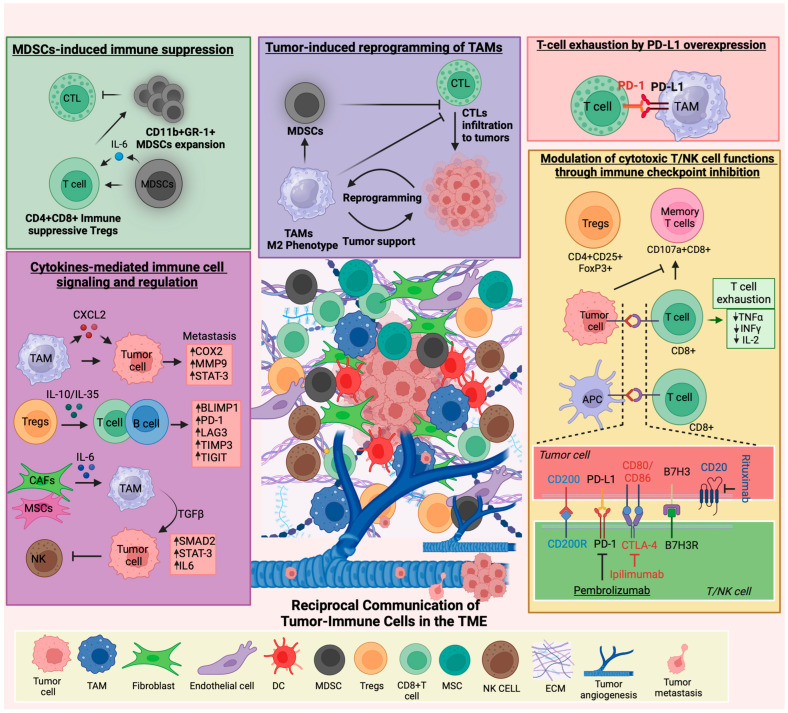
Dynamic tumor-immune cell interactions in the pediatric TME. The TME encompasses tumor cells alongside recruited immune cells, endothelial cells, fibroblasts, adipocytes, and non-cellular elements of the extracellular matrix. Over time, myeloid-derived suppressor cells (MDSCs), mesenchymal stromal cells (MSCs), tumor-associated macrophages (TAMs), T cells, and fibroblasts undergo education by the tumor, leading to a transition toward immune-suppressive and pro-tumorigenic functions. For instance, TAMs reprogramming toward pro-tumorigenic phenotypes by tumor cells promote myeloid-derived suppressor cell (MDSC) infiltration and proliferation, resulting in the inhibition of CTL infiltration into tumors and the upregulation of CD4+CD8+ immune-suppressive Tregs. These Tregs further support MDSC expansion, promoting diverse phenotypes within the primary tumor, including growth and metastasis. Tumor-educated immune cells secrete a plethora of pro-tumorigenic cytokines and growth factors that influence signaling in the tumor and neighboring immune cells. For example, chemokine (C-X-C motif) ligand 2 (CXCL2) secreted by TAMs upregulates cyclooxygenase-2 (COX2), matrix metalloproteinase-9 (MMP9), and STAT-3 in tumor cells, promoting tumor metastasis. IL-10 and IL-35 secretions from Tregs elevate the expression of inhibitory molecules like B lymphocyte-induced maturation protein-1 (BLIMP1), programmed cell death protein 1 (PD-1), lymphocyte-activation gene 3 (LAG3), metalloproteinase inhibitor 3 (TIMP3), and TIGIT on T and B cells, inhibiting their proliferation, activation, and tumor cytotoxicity. Similarly, cancer-associated fibroblasts (CAFs) and MSCs secrete IL-6, which increases the production of TGFβ in the TME. TGF-β1 induces the expression of IL-6 in tumor cells, promoting STAT-3 and SMAD activation, fostering the survival of TAMs, and inhibiting NK cell functions. Elevated expression of immune checkpoint molecules in tumors or antigen-presenting cells (APCs) like macrophages or dendritic cells (DCs) hinders the activation of cytotoxic NK cells and CD8+ T cells. This inhibition restricts anti-tumor responses by curtailing the activation, expansion, and infiltration of T and NK cells. Crucial to these regulatory processes is the upregulation of immune checkpoint molecules like CD200, PD-L1, CD80, B7H3, and CD20 on tumor cells, which play pivotal roles in modulating immune responses. Interactions of these molecules with their receptors on T or NK cells, such as CD200-CD200R, PD-L1-PD-1, and CD80-CTLA-4, B7-H3, and its receptor, contribute to immune suppression by inhibiting T and NK cell activation. These interactions underscore the intricate mechanisms that tumors and the immune system employ to regulate immune responses and underscore their potential as therapeutic targets in pediatric tumors. Pembrolizumab (anti-PD-1) and ipilimumab (anti-CTLA-4) have gained approval for treating pediatric cancers by modulating immune responses. Additionally, rituximab, targeting CD20, is utilized for B cell depletion in conditions such as lymphomas. These advancements in immunotherapy exemplify the progress made in treating specific cancers by targeting immune checkpoints or cell surface antigens.

**Table 1 cancers-16-02201-t001:** Cytokines in the pediatric tumor microenvironment and their roles and potential as predictive biomarkers.

Cytokines and Their Receptors	Tumor Type	Cell Source	Prognosis and Correlation with Treatment	Role in Tumor Progression	Mechanisms
**Hematological malignancies:**					
TNFα [90,98]	AML	Blast cells, LSC, TAMs	Worse OS	Promoting	Promotes tumor inflammation by activating NF-kB, PI3K/AKT, and MAPK signaling
IL3Rα (CD123) [99,100]	B-ALL	Blast cells	Good OS	Promoting	Alters CXCR4/SDF-1 interaction in the BM, promotes AML cell growth
IL-7Rα [101,102,103]	T-ALL	T cells	Worse OS	Promoting	Promotes JAK, STAT-5, PI3K signaling
IL-2 [104,105,106,107]	AML	T cells	Good OS	Inhibitory	Stimulates CTLs production
IL-6 [108,109]	AML	Blast cells, BMSCs	Worse OS	Promoting	IL-6-induced STAT3 pathway promotes AML progression
IL-10 [110,111,112,113]	ALL	Myeloid, lymphoid cells	Good OS	Inhibitory	IL-10 family cytokines maintain tissue balance by regulating inflammation, enhancing immunity, and aiding tissue repair. Children with ALL have low IL-10 levels. IL-10 inhibits AML blast proliferation, while its deficiency impairs B-cell development, increases DNA damage, and raises proinflammatory cytokines IL-1α, IL-6, IL-12p40, IL-13, MIP-1β/CCL4, and G-CSF.
IL-33, IL1RL1 [114,115,116]	AML	AML, BM, mast, ILC2, Tregs and TAM cells	Worse OS	Promoting	Activates p38 MAPK, Wnt, and Notch pathways, promoting cell survival and stemness
IL1β [92,117,118,119]	AML/ALL	Leukemia cells, monocytes	Worse OS	Promoting	Activates IL-1/p38MAPK pathway and promotes leukemia progression
**Solid tumors:**					
IL-1β and TNF-α [120]	NB	TAMs	Worse OS	Promoting	Promotes ARG2 expression via p38/ERK signaling
IL-2 [121,122,123,124]	NB	T and NK cells	Good OS	Inhibitory	promote CD8+ T, B, and NK cell cytotoxicity activity
IL-6 [125,126]	NB	BMSCs, MSC, CAFs, NB cells	Worse OS	Promoting	Promotes STAT-3, ERK signaling, inhibits NK functions
IL-6 [87,127,128]	Glioma	Glial, stromal, or TME cells	Worse OS	Promoting	Promotes oncogenic JAK/STAT3 signaling
IL-8 [129,130,131,132]	Osteosarcoma	Osteosarcoma cells, MSC, TAMs	Worse OS	Promoting	Promotes FAK signaling, ABCB1/MDR1 pathway and immune suppression
IL-10 [87,133,134]	MB, GCT, Glioma	Bregs	Worse OS	Promoting	IL-10 suppresses T and NK cell anti-tumor immunity and promotes cell proliferation.
IL-6, IL-10 [87,135,136]	MB	MB cells	Worse OS	Promoting	IL-6 induces STAT3 pathway and promotes MB growth
IL-12 [137,138,139]	Osteosarcoma	APCs	Good OS	Inhibiting	Activates NK, CTLs, and memory T cells, inhibiting tumor growth and metastasis
IL-13Rα2 [140,141,142]	Glioma	Glioma cells	Worse OS	Promoting	IL-13 and its receptor IL-13Rα2 signals through JAK/STAT and AP-1 pathways
TGF-β [126,143]	NB	CAFs, TAMs	Worse OS	Promoting	Activates TGF-β/IL-6 pathway in NB and MSC, promotes immune suppression
VEGF [144,145,146,147]	ES	ES cells	Worse OS	Promoting	Attracts BM-derived EPCs to drive tumor blood vessel growth
VEGF [148,149,150]	WT	WT and stromal cells	Worse OS	Promoting	Regulates angiogenesis by activating VEGFR-1 and VEGFR-2
AFP [151,152,153,154]	Hepato blastoma	Liver cancer cells	Worse OS	Promoting	Promotes PI3K/AKT, suppresses Fas/FADD apoptotic pathway

AML: Acute myeloid leukemia, B-ALL: B-cell acute lymphoblastic leukemia, T-ALL: T-cell acute lymphoblastic leukemia, ALL: Acute lymphoblastic leukemia, NB: Neuroblastoma, MB: Medulloblastoma, GCT: Germ Cell Tumors, ES: Ewing Sarcoma, WT: Wilms tumor. OS: Overall survival, TNFα: Tumor necrosis factor α, LSC: Leukemia stem cells, TAM: Tumor-associated macrophages, IL: Interleukin, IL-3RA: Interleukin-3 Receptor Alpha Chain, SDF-1: Stromal-derived factor-1, CTLs: Cytotoxic T lymphocytes, MIP-1β/CCL4: Macrophage inflammatory protein-1β/CCL4, G-CSF: Granulocyte colony-stimulating factor, BM: Bone marrow, ILC2s: Group 2 innate lymphoid cells, Tregs: T regulatory cells, IL1RL1: Interleukin 1 receptor-like 1, ARG2: arginase 2, BMSCs: Bone marrow stromal cells, MSC: Mesenchymal stromal cells, CAF: Cancer-associated fibroblasts, IL-13Rα2: Interleukin-13 receptor alpha 2, Bregs: B regulatory cells (CD19+ IL-10+ or CD19+ CD24hiCD38hi), APC: Antigen-presenting cells, VEGF: Vascular endothelial growth factor, EPCs: Endothelial progenitor cells, VEGFR: Vascular endothelial growth factor receptor, AFP: Alpha-fetoprotein.

**Table 2 cancers-16-02201-t002:** Clinical trials investigating the use of targeted mAbs and adoptive T-cell therapies, either in combination with standard cancer treatments or as single agents, for the treatment of pediatric cancers.

Cancer Type	NCT; Phase	Monoclonal Antibodies or Immune Cells	Target	Combination Drugs	Status and Outcome
**Hematological malignancies:**					
Relapsed/Refractory Mature B-NHL	NCT05533775; I/II	Glofitamab	CD20	Rituximab, Ifosfamide, Carboplatin, Etoposide	Recruiting
Ph-like ALL	NCT03571321; I	Ruxolitinib	JAK1 JAK2	Rituximab, Cyclophosphamide, Cytarabine, Mercaptopurine, Vincristine, Pegaspargase, Methotrexate, dexamethasone, Doxorubicin, Thioguanine	Recruiting
Relapsed B-ALL	NCT05645718; II	Inotuzumab Ozogamicin, Blinatumomab and Rituximab	Tumor cells	Cyclophosphamide, Vincristine, and Dexamethasone	Recruiting
Aggressive B-cell Lymphoma	NCT03864419; I	Rituximab Hyaluronidase	CD20	Cyclophosphamide, Vincristine, Methotrexate, Etoposide, Doxorubicin, Prednisone	Completed
B-ALL	NCT03150693; III	Inotuzumab Ozogamicin	CD22	Rituximab, Allopurinol, Cytarabine, Daunorubicin, Vincristine, Dexamethasone, PLA, Methotrexate, Cyclophosphamide, Mercaptopurine, Doxorubicin, Thioguanine	Suspended due to high toxicity
Relapsed/Refractory HL [190]	NCT01896999; I/II	Brentuximab vedotin	CD30	Nivolumab and Ipilimumab	CRR improved, Suspended
PD-L1+ve solid tumors, lymphoma [59]	NCT02332668; I/II	Pembrolizumab	PD-1	Single agent	Recruiting
Relapsed/Refractory B-ALL or B-NHL	NCT04544592; I/II NCT04173988; I	CAR-T cells	CD19	Single agent	Recruiting Not Recruiting
Relapsed/Refractory B-ALL [191]	NCT02650414; I/II	CAR-T cells	CD22	Single agent	Recruiting
Relapsed/Refractory HL	NCT04268706; II	CAR-T cells	CD30	Single agent	Recruiting
Relapsed/Refractory B-ALL, B-NHL	NCT03743246; I/II	JCAR017 (CAR-T cells)	CD19	Fludarabine, Cyclophosphamide	Completed
Relapsed/ Refractory pre-B ALL	NCT03605589; I	Pembrolizumab	PD-1	blinatumomab	Withdrawn due to low enrollment
**Solid tumors:**					
Relapsed/ Refractory NB	NCT04238819; 1b/2	Abemaciclib	CDK4, CDK6	Dinutuximab, GM-CSF, Irinotecan, Temozolomide	Active, Not recruiting
	NCT03794349; II	Eflornithine	polyamines	Irinotecan, Temozolomide, Dinutuximab	Recruiting
	NCT02914405; I	Dinutuximab	GD2	mlBG, Nivolumab (anti-PD1) antibody	Recruiting
	NCT05400603; I [192]	Allogeneic γδ T Cells	Tumor cells	Temozolomide, Irinotecan, Dinutuximab, Zoledronate	Recruiting
INI1(-) or SMARCA4-def. tumors	NCT05407441; I/II	tazemetostat	EZH2	Nivolumab and Ipilimumab	Recruiting
Recurrent or Progressive HGG	NCT04323046; I	nivolumab	PD-1	Single agent. Given before and after surgery	Recruiting
High-grade primary CNS malignancies [193]	NCT03130959; 1b/II	Nivolumab	PD-1	Ipilimumab	No clinical benefit in combination
Relapsed/Refractory solid tumors	NCT05302921; II	Nivolumab, Ipilimumab	PD-1, CTLA-4	cryoablation therapy	Active, not recruiting
Relapsed, Refractory, or Progressive CNS solid tumors and lymphomas [194]	NCT03445858; I	Pembrolizumab	PD-1	Decitabine and radiation therapy	Active, not recruiting
Refractory gliomas, MB	NCT02359565; I	Pembrolizumab	PD-1	Single agent	Recruiting
Liver cancer [195]	NCT04134559; II	Pembrolizumab	PD-1	Single agent	Recruiting
GD2+ve Brain tumors	NCT04099797; I	C7R-GD2.CAR T Cells	GD2	Single agent	Recruiting

NHL: Non-Hodgkin Lymphoma, ALL: Acute Lymphoblastic Leukemia, B-ALL: B-Cell Acute Lymphoblastic Leukemia, HL: Hodgkin Lymphoma, B-NHL: B-cell Non-Hodgkin Lymphoma, NB: Neuroblastoma, INI1(-): INI1 negative, SMARCA4-def: SMARCA4 deficient, HGG: High-Grade Glioma, CNS: central nervous system, MB: medulloblastoma, PLA: Pegylated L-Asparaginase, EZH2: Enhancer of zeste homolog 2, JAK: Janus kinase, mlBG: 131-l Metaiodobenzylguanidine, CRR: complete response rate.

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
