# Peer review of "Immune Microenvironment in Childhood Cancers: Characteristics and Therapeutic Challenges"

_cancers, 2024, doi:10.3390/cancers16122201_

Round 1

Reviewer 1 Report

Comments and Suggestions for Authors

The topic is interesting, the exposition is clear, the references are up to date. The conclusions are also good. I suggest the author expand the chapter on immunotherapy.

 Add a table with the abbreviations used in the text.

Minor Point: Page 6, Lines 31,32: "chemotherapy can induce changes in the TME of retinoblastoma, transforming it into a more immunogenic and immune-infiltrated environment" ??? Previously, the author has claimed the opposite.

Author Response

Comments and Suggestions for Authors

The topic is interesting, the exposition is clear, the references are up to date. The conclusions are also good. I suggest the author expand the chapter on immunotherapy.

Reply: I appreciate the author's thorough review of the manuscript and the insightful comments provided. Following reviewer suggestions, the chapter on immunotherapy has been expanded in the revised manuscript. 

Add a table with the abbreviations used in the text.

Reply: As suggested, the revised manuscript now includes a table containing abbreviations used throughout the text.

Minor Point: Page 6, Lines 31,32: "chemotherapy can induce changes in the TME of retinoblastoma, transforming it into a more immunogenic and immune-infiltrated environment" ??? Previously, the author has claimed the opposite.

Reply: This discrepancy has been rectified in the revised manuscript. The actual message conveyed by these studies is that the overexpression of PD-1L/PD-1I or other immune checkpoint molecules creates a tumor-suppressive microenvironment.

Reviewer 2 Report

Comments and Suggestions for Authors

The review article entitled “Immune microenvironment in childhood cancers: Characteristics and therapeutic challenges” by Anup Singh Pathania is focused on pediatric cancers, both hematological malignancies and solid tumors, and is aimed at describing the pediatric immune tumor microenvironment and its implications in pediatric cancer therapies.

The issue is of interest, but a more accurate, and better organized description is recommended.

In the introduction, please provide a more comprehensive but concise picture of pediatric cancers, by distinguishing clearly between hematological malignancies and solid tumors. Please specify incidence and outcome of pediatric tumors compared to adults.

Additional relevant points:

1)      Accurate English revision should be done throughout the paper.

2)      Make the manuscript more concise by avoiding repetition of concepts.

3)      Some sentences such as: …”makes these tumors less immuno genic and more immune tolerant” can be misleading since a tumor is not “immune tolerant”.

4)      Some sentences are unclear and need to be reformulated such as: “Among CNS tumors, pediatric gliomas exhibit the highest number of T-cell receptor (TCR) clones, whereas extracranial tumors such as osteosarcoma and neuroblastoma show the highest TCR diversity”. Or: “…in osteosarcoma patients diagnosed with primary metastasis”…

5)      Abbreviation should be provided at the first use of a word and then used throughout the manuscript.

6)      Appropriate simple summary and abstract should be provided.

7)      Table 1. Please group tumors in hematological malignancies and solid tumors. Please unify columns “Prognosis” and “Correlation with treatment”. In the table, the role of IL10 deficiency in ALL onset should be better explained. In addition, the promoting role of IL10 in other tumors should be included.

8)      Table 2. Please insert table 2 after it is cited in the main text. Please group tumors in hematological malignancies and solid tumors. The column “Drug” should be changed in “monoclonal antibodies or immune effectors” and in the column “combination drugs” please check the appropriate content. Whenever possible insert appropriate references related to clinical trials.

9)      Clearly indicate FDA approved immunotherapies for pediatric cancers.

10)   Include a point by point conclusion paragraph adding the issue of patient selection for a better outcome of immunotherapy or targeted therapies.

Comments on the Quality of English Language

1Accurate English revision is needed throughout the paper

Author Response

Comments and Suggestions for Authors

The review article entitled “Immune microenvironment in childhood cancers: Characteristics and therapeutic challenges” by Anup Singh Pathania is focused on pediatric cancers, both hematological malignancies and solid tumors, and is aimed at describing the pediatric immune tumor microenvironment and its implications in pediatric cancer therapies.

The issue is of interest, but a more accurate, and better organized description is recommended.

In the introduction, please provide a more comprehensive but concise picture of pediatric cancers, by distinguishing clearly between hematological malignancies and solid tumors. Please specify incidence and outcome of pediatric tumors compared to adults.

Reply: I sincerely appreciate the reviewer for their constructive review and invaluable suggestions, which have helped identify errors and enhance the manuscript. Following the recommendations, I have thoroughly revised the manuscript to provide a more comprehensive picture of pediatric hematological versus solid malignancies. Additionally, statistics on pediatric tumors, as well as the incidence and outcomes of pediatric tumors compared to adults, have been included in the revised manuscript.

Additional relevant points:

1)      Accurate English revision should be done throughout the paper.

        Reply: The manuscript has been proofread by native English speakers.

 2)      Make the manuscript more concise by avoiding repetition of concepts.

       Reply: The repetitive concepts have been eliminated, and redundancy has been reduced.

3)      Some sentences such as: …”makes these tumors less immuno genic and more immune tolerant” can be misleading since a tumor is not “immune tolerant”.

      Reply: Misleading sentences have been removed.

4)      Some sentences are unclear and need to be reformulated such as: “Among CNS tumors, pediatric gliomas exhibit the highest number of T-cell receptor (TCR) clones, whereas extracranial tumors such as osteosarcoma and neuroblastoma show the highest TCR diversity”. Or: “…in osteosarcoma patients diagnosed with primary metastasis”…

      Reply: The following sentence and several others have been rewritten to better convey the message.

5)      Abbreviation should be provided at the first use of a word and then used throughout the manuscript.

     Reply: Abbreviations are provided at the first use

 6)      Appropriate simple summary and abstract should be provided.

      Reply: A simple summary and abstract have been included in the revised manuscript.

 7)      Table 1. Please group tumors in hematological malignancies and solid tumors. Please unify columns “Prognosis” and “Correlation with treatment”. In the table, the role of IL10 deficiency in ALL onset should be better explained. In addition, the promoting role of IL10 in other tumors should be included.

      Reply: In the revised manuscript, the "Prognosis" and "Correlation with Treatment" columns in Table 1 have been unified. Additionally, the role of IL-10 deficiency in the onset of acute lymphoblastic leukemia (ALL) is explained in greater detail, and the tumor-promoting role of IL-10 has been included.

       8)      Table 2. Please insert table 2 after it is cited in the main text. Please group tumors in hematological malignancies and solid tumors. The column “Drug” should be changed in “monoclonal antibodies or immune effectors” and in the column “combination drugs” please check the appropriate content. Whenever possible insert appropriate references related to clinical trials.

       Reply: In the revised manuscript, Table 2 has been updated to group tumors into hematological malignancies and solid tumors. The term "Drug" has been changed to "monoclonal antibodies" or "immune effectors." The entire table has been rechecked. Most of the clinical trials mentioned are still underway, with no published papers available yet. However, a few trials have been reported in publications, which are cited in the revised manuscript.

9)      Clearly indicate FDA approved immunotherapies for pediatric cancers.

       Reply:  So far, these monoclonal antibodies have been approved for pediatric cancers: blinatumomab for pediatric patients with CD19-positive B-cell precursor acute lymphoblastic leukemia (ALL); dinutuximab, an anti-GD2 mAb, for the treatment of neuroblastoma; rituximab, an anti-CD20 mAb, for ALL; ipilimumab, an anti-CTLA-4 antibody, for pediatric patients with advanced melanoma; and pembrolizumab, an anti-PD-1 antibody, approved for specific subsets of pediatric patients. These FDA-approved immunotherapies have been thoroughly discussed in the revised manuscript under section 3.4: Immunotherapy Response and Predictors in Pediatric Tumors.

10)   Include a point by point conclusion paragraph adding the issue of patient selection for a better outcome of immunotherapy or targeted therapies.

Reply: This comment has been addressed in section 3.4: Immunotherapy Response and Predictors in Pediatric Tumors.

Reviewer 3 Report

Comments and Suggestions for Authors

The article presents a comprehensive and systematic review of the complex subject regarding the profiling of the immune microenvironment in various pediatric tumors, its impact on tumor development, metastasis, and response to therapeutic modalities. In contrast to adult cancers, there is much less knowledge about the microenvironment of children's cancers.

The use of molecular target-based treatments and/or immunotherapy is crucial for improving the outcome of pediatric cancers. By gaining a more profound understanding of their microenvironment and their relationship with the immune system, it may be possible to create more effective immunotherapies.

This article is interesting because it approaches the issue from the perspective of individual biological processes, rather than focusing on specific cancers.

Although I have a positive view of sections 1, 2 and 3.1-3.3, I find section 3.4 on the response to and predictors of immunotherapy in pediatric tumors to be too general in its current form. The Author lists the possibilities of immunotherapy in pediatric cancers, but the subsequent discussion is primarily based on existing review articles, lacking a comprehensive review of primary research studies.

Additionally, there are some minor comments:

- Section 2: Profiling the immune microenvironment of pediatric tumors - Define TAM abbreviation when first used

- Figure 1 - in the figure description there should be expansions of abbreviations: MDSCs, MSCs, TAMs

-conclusions: “Encouraging preclinical investigations… “  - supply key references for examples of such investigations

Author Response

Comments and Suggestions for Authors

The article presents a comprehensive and systematic review of the complex subject regarding the profiling of the immune microenvironment in various pediatric tumors, its impact on tumor development, metastasis, and response to therapeutic modalities. In contrast to adult cancers, there is much less knowledge about the microenvironment of children's cancers.

The use of molecular target-based treatments and/or immunotherapy is crucial for improving the outcome of pediatric cancers. By gaining a more profound understanding of their microenvironment and their relationship with the immune system, it may be possible to create more effective immunotherapies.

This article is interesting because it approaches the issue from the perspective of individual biological processes, rather than focusing on specific cancers.

Although I have a positive view of sections 1, 2 and 3.1-3.3, I find section 3.4 on the response to and predictors of immunotherapy in pediatric tumors to be too general in its current form. The Author lists the possibilities of immunotherapy in pediatric cancers, but the subsequent discussion is primarily based on existing review articles, lacking a comprehensive review of primary research studies.

Reply: I am grateful to the reviewer for their constructive feedback and invaluable suggestions, which have greatly contributed to improving the manuscript. As suggested, section 3.4 on Immunotherapy response and predictors in pediatric tumors has been thoroughly revised, with the studies discussed in more detail.

Additionally, there are some minor comments:

  • Section 2: Profiling the immune microenvironment of pediatric tumors - Define TAM abbreviation when first used.
    Reply: TAM abbreviation was defined in the first use in revised manuscript.

- Figure 1 - in the figure description there should be expansions of abbreviations: MDSCs, MSCs, TAMs

Reply: All abbreviations have been defined in the figure description.

-conclusions: “Encouraging preclinical investigations… “  - supply key references for examples of such investigations

Reply: References have been included in the revised manuscript.

Round 2

Reviewer 2 Report

Comments and Suggestions for Authors

The revised manuscript was significantly improved and deserves publication

Reviewer 3 Report

Comments and Suggestions for Authors

none